# Exploration of the factors related to self-efficacy among psychiatric nurses

**Hironori Yada** [1]*, **Hiroshi Abe**[2], **Ryo Odachi**[1], **Keiichiro Adachi**[3]

**1** Department of Clinical Nursing, Yamaguchi University Graduate School of Medicine, Yamaguchi, Japan,
**2** Department of Clinical Psychology, Health Sciences University of Hokkaido, Hokkaido, Japan,
**3** Department of Fundamental Nursing, Yamaguchi University Graduate School of Medicine, Yamaguchi, Japan

* yadahiro@yamaguchi-u.ac.jp

**Data Availability Statement:** All relevant data are within the Supporting Information file.

**Funding:** This work was supported by JSPS KAKENHI(https://www.jsps.go.jp/j-grantsinaid/) [grant number 19K19498]. The supporting source

## Abstract

The average length of hospital stay in the psychiatric ward is longer, and the risk of patient-to-nurse violence is higher than that in other departments. Therefore, psychiatric nurses' work environment may differ from that of other nurses. The factors related to psychiatric nurses' self-efficacy may also differ from those of general workers or other nurses. Mental health care that considers the characteristics of psychiatric nurses requires exploration of self-efficacy unique to psychiatric nurses. This cross-sectional study aimed to explore the distinct factors related to psychiatric nurses' self-efficacy. The developed 24 items related to improvement in self-efficacy and 25 items related to decrease in self-efficacy were examined. The Generalized Self-Efficacy Scale was used to measure the validity of the factors. To extract the factors of self-efficacy, data from 132 nurses and assistant nurses who provided informed consent were analyzed, and the reliability and validity of the factors were calculated. The factors associated with improvement in self-efficacy were "Positive reactions by patients," "Ability to positively change nurse-patient relationship," and "Practicability of appropriate nursing." The factors associated with decrease in self-efficacy were "Uncertainty in psychiatric nursing" and "Nurses' role loss." The Cronbach's α for all factors exceeded .70. Of the five factors, four had significant weak-to-moderate correlations with the Japanese version of the Generalized Self-Efficacy Scale; therefore, the validity was quantitatively confirmed with four factors. Interventions based on these four factors may improve psychiatric nurses' self-efficacy. Additionally, it is possible that this tool assesses the unique facets of self-efficacy rather than psychiatric nurses' general self-efficacy. Interventions to improve psychiatric nurses' self-efficacy based on the characteristics of psychiatry are needed.

## Introduction

Albert Bandura defined self-efficacy as "judgments of how well one can execute courses of action required to deal with prospective situations" (p. 122) [1]. People with high self-efficacy set goals to challenge and improve their task achievement rate; however, people with low self-

had no involvements for this study design (collection, analysis, and interpretation of data), writing the report and the decision to submit the report for publication.

**Competing interests:** The authors have declared that no competing interests exist.

efficacy tend to have fluctuation in their ways of thinking, which results in dampened spirits [2]. Self-efficacy affects mental health [3]. Additionally, psychiatric nurses experience lower mental health levels than do regular nurses [4].

Workers' self-efficacy is often influenced by their performance [5]. Concerning nurses, it is difficult to set clear numerical values of past achievements, because they cannot be measured monetarily, unlike business outcomes. In addition, psychiatric nurses specialize in caring for patients with mental disorders, who are known to have high recurrence rates [6]. In 2016, patients' average hospital stay in general wards was 16.2 days, while that in psychiatric wards was 269.9 days in Japan [7]. Furthermore, the average length of stay at Japanese psychiatric hospitals is significantly longer than that in other Organization for Economic Cooperation and Development (OECD) countries [8]. Therefore, psychiatric nurses' achievements that affect their self-efficacy may differ from those of general workers or other nurses. In such a situation, psychiatric nurses feel that uncertainty of care and an unmotivated appearance of the patient can lead to reduced self-efficacy [9]. Consequently, nurses are likely to give up active involvement with patients who will not be leaving the hospital [10].

Furthermore, psychiatric nursing is a stressful profession [11], and is associated with depression [12]. There is, thus, the concern that psychiatric nurse's mental health may deteriorate after they work in psychiatric wards for a long period of time. More than half of the psychiatric nurses who have experienced violence have experienced depressive symptoms [13]. Stress is related to nurses' self-efficacy [14], which, among psychiatric nurses, may be low. Self-efficacy is an important factor affecting mental health, as well as stress [3]. Nonetheless, previous studies have not quantitatively clarified the self-efficacy in psychiatric nurses. A qualitative study identified 18 factors influencing self-efficacy among psychiatric nurses, such as "the patients presenting a positive attitude," "building a relationship of trust with the patients," "uncertainty in caregiving," and "sense of loss regarding one's role as a nurse" [9]. Thus, a quantitative analysis of these factors is required, which may shed light on ways to improve nurses' mental health and motivation to work as well as reduce their turnover intention. Moreover, a better understanding of psychiatric nurses' self-efficacy would contribute to a concrete mental health care planning. Thus, this cross-sectional study aimed to explore the distinct factors related to psychiatric nurses' self-efficacy.

## Methods

### Participants and procedure

In this study, the criterion for including participants was nurses and assistant nurses in the psychiatric ward, and exclusion criterion was nurses and assistant nurses in wards other than the psychiatric ward. The study participants were 147 registered nurses and assistant nurses working at public and private psychiatric hospitals in a prefecture of Japan. The principal researcher requested the cooperation of nursing directors at each hospital, in writing and verbally, and anonymous self-administered questionnaires were distributed and recovered by nursing managers in September 2016. Questionnaires were distributed to individual participants, and their responses were collected through the nursing directors. The participants provided written informed consent, could freely decline participation in the survey, and were neither compensated nor rewarded for participating. Each participant was given an envelope, and the questionnaires were sealed. Therefore, the privacy of participants was protected. The survey was anonymous, and only the researcher could access the data. This study was approved by the Ethics Review Board of Yamaguchi University Graduate School of Medicine, School of Health Sciences (approval no. 400).

## Measures

General demographic data including age, years of experience as a nurse, sex, nursing education level, and job position were collected. The scale (The Factors Related to Improved Self-efficacy) included 24 items assessing "improved self-efficacy" and the scale (The Factors Related to Decreased Self-efficacy) included 25 items assessing "decreased self-efficacy" were developed based on previously determined qualitative data [9], and items were rated on a 5-point Likert scale from 1 (*I do not think so at all*) to 5 (*I think so*). For each question in the Factors Related to Improved Self-efficacy scale, higher scores, indicated by the response "I think so," indicate a high self-efficacy. For each question in the Factors Related to Decreased Self-efficacy scale, higher scores, indicated by the response "I think so," indicate a low self-efficacy. Three researchers with experience as a psychiatric nurse comprehensively reviewed the qualitative data covering the self-efficacy of psychiatric nurses [9] and created the question items. The content and face validity were confirmed through consensus.

Additionally, the Generalized Self-Efficacy Scale (GSES)[15] that was translated to Japanese (GSES-J)[16] was used to evaluate the validity of the factors related to self-efficacy. The reliability and validity of the GSES [15,17] and GSES-J [16] has been established. The GSES-J comprises one factor with 23 items, which are rated on a 5-point Likert scale ranging from 1 (*I do not think so*) to 5 (*I think so*). For each question, higher scores, indicated by the response "I think so," indicate a high self-efficacy. The Cronbach's α coefficient in this study was .88, confirming its reliability as a one-factor structure.

## Sample size

Sample size needs more than 5 times the number of items when exploratory factor analysis (EFA) is the main analysis [18]. There are 24 items assessing improved self-efficacy and 25 items assessing reduced self-efficacy. Therefore, the minimum sample size required was 120–125. Our sample had 125 respondents.

## Statistical analyses

Means, standard deviations (*SD*), and frequency (*n*) were calculated for participants' demographic characteristics. Ceiling effect, floor effect, kurtosis, and skewness of the items were confirmed by observing their distribution on the item scores of the factors that improve and decrease self-efficacy. The factor structure was identified using EFA. The internal consistency of the factors was calculated using Cronbach's alpha coefficient. Pearson's correlation coefficients were calculated to confirm the correlation between each factor and the GSES-J. The significance level was set at $p < .05$. IBM SPSS 24.0 was used for all analyses (Windows; SPSS, Chicago, IL, USA).

## Results

### Demographics

Responses were received from 147 participants. Data from 132 participants who provided written informed consent were analyzed (effective response rate = 89.7%). Participants' mean age was 39.73 years (*SD* = 9.75; range = 21.00–64.00; 2 missing). The mean years of nursing experience was 13.05 years (*SD* = 10.57; range = 0.00–40.00; 3 missing). There were 58 men (43.9%), 73 women (55.3%), and 1 left this unanswered (0.8%). There were 11 university graduates (8.3%), 2 junior nursing college graduates (1.5%), 112 nursing school graduates (84.8%), 6 with another level of education (4.5%), and 1 left this unanswered (0.8%). Regarding job positions, 22 were managers (16.7%), 104 were non-managers (78.8%), and 6 left this unanswered (4.5%). Table 1 presents the demographic characteristics of participants.

**Table 1. Demographic details of participants.**

| Variable | Mean or Number | Standard deviation or percentage |
|---|---|---|
| Mean age (years) | 39.73 | 9.75 |
| Mean nursing experience (years) | 13.05 | 10.57 |
| Sex | | |
| Male | 58 | 43.9% |
| Female | 73 | 55.3% |
| Unanswered | 1 | 0.8 |
| Educational background | | |
| University | 11 | 8.3% |
| Junior nursing college | 2 | 1.5% |
| Nursing school | 112 | 84.8% |
| Another level of education | 6 | 4.5% |
| Unanswered | 1 | 0.8% |
| Job positions | | |
| Manager | 22 | 16.7% |
| Non-manager | 104 | 78.8% |
| Unanswered | 6 | 4.5% |

## Factors related to self-efficacy among psychiatric nurses

The number of missing values in each item ranged from 0–4, which were judged as small for the analysis. Thus, the mean value was substituted for the missing data in the statistical analyses. Normal distribution was assumed, because the skewness and kurtosis values did not exceed ±2 [19]. Three items related to decreased self-efficacy had a ceiling effect at + 1 SD, and these were excluded from the analyses. In the EFA, the maximum likelihood method was used for factor extraction; the Kaiser-Guttmann criterion was also used for determining the number of factors. In the EFA, the Kaiser-Meyer-Olkin (KMO) measure of sampling adequacy and Bartlett's test of sphericity ($\chi^2$) were used. The required sample size was determined using the KMO measure of sampling adequacy. The appropriateness of factor analysis was assessed using Bartlett's test of sphericity ($\chi^2$). Items that showed a factor loading of less than 0.40 on one factor or greater than 0.40 on multiple factors were deleted. After the items were deleted, the EFA was repeated. The results revealed that three factors (13 items) that focused on improved self-efficacy (Table 2) and two factors (12 items) related to decreased self-efficacy (Table 3) were extracted using factor analysis. Among the factors related to improved self-efficacy (Fig 1), 3 factors had an eigenvalue of 1 or more (eigenvalues: 5.51, 1.45, and 1.16), and 2 factors (eigenvalues: 4.20 and 2.53) were related to decreased self-efficacy (Fig 2). The variance of the three factors in Fig 1 was a cumulative of 51.38%. The variance of the two factors in Fig 2 was a cumulative of 47.73%.

During extraction, the KMO measure of sampling adequacy was 0.88, and Bartlett's test was $\chi^2$ (df) = 704.10 (78) ($p < .001$). The first factor comprised items related to positive emotions experienced by nurses triggered by patients, which was named "Positive reactions of patients." The second factor comprised items related to the "Ability to positively change nurse-patient relationship." The third factor comprised items related to whether a nursing action or procedure was considered appropriate and feasible by nurses, which was named "Practicability of appropriate nursing."

During extraction of the factor that decreases self-efficacy, the KMO measure of sampling adequacy was 0.83 and Bartlett's test was $\chi^2$ (df) = 596.23 (66) ($p < .001$). The first factor comprised items related to patients' responses that deviated from nursing expectations, which was named "Uncertainty in psychiatric nursing." The second factor comprised items related to

**Table 2. Factors related to improved self-efficacy among psychiatric nurses (n = 132).**

| Item no. | Item content | Factor 1 | Factor 2 | Factor 3 |
|---|---|---|---|---|
| Factor 1: Positive reactions of patients (Cronbach's α = 0.80) | | | | |
| 15 | Patients express their gratitude | **0.88** | -0.09 | -0.02 |
| 14 | By being considerate to patients, I feel that they also express their gratitude in return | **0.69** | -0.02 | -0.01 |
| 12 | By being considerate to patients, I can see their smiling faces | **0.58** | 0.06 | 0.10 |
| 16 | I feel that patients who are usually negative open their hearts to me | **0.50** | 0.18 | 0.16 |
| Factor 2: Ability to positively change nurse-patient relationship (Cronbach's α: 0.84) | | | | |
| 18 | I can get patients to understand by explaining things | -0.09 | **0.89** | -0.08 |
| 19 | I can build trusting relationships with patients | 0.24 | **0.81** | -0.18 |
| 21 | I can make patients happy | 0.23 | **0.58** | 0.04 |
| 17 | I can improve patients' opinions about rehabilitation | 0.02 | **0.51** | 0.35 |
| 22 | I feel that, among nurses of the same rank, I am relied upon | -0.23 | **0.45** | 0.35 |
| Factor 3: Practicability of appropriate nursing (Cronbach's α coefficient: 0.71) | | | | |
| 2 | I can anticipate symptoms and care | 0.13 | -0.16 | **0.75** |
| 4 | I can practice nursing that I think is correct | -0.12 | 0.21 | **0.59** |
| 2 | I can make use of training | 0.04 | -0.04 | **0.56** |
| 6 | I feel patients' recovery goes as anticipated | 0.10 | -0.06 | **0.53** |
| | Factor correlations | | | |
| | Factor 1 | -- | | |
| | Factor 2 | .60 | -- | |
| | Factor 3 | .49 | .64 | -- |

nurses experiencing a sense of loss regarding their role and perceived distance from the patients, called "Nurses' role loss."

## Reliability of factors

Concerning reliability, for all factors, the Cronbach's α exceeded .70 (Table 4).

**Table 3. Factors related to decreased self-efficacy among psychiatric nurses (n = 132).**

| Item no. | Item content | Factor 1 | Factor 2 |
|---|---|---|---|
| Factor 1: Uncertainty in psychiatric nursing (Cronbach's α: 0.86) | | | |
| 3 | I have been refused by patients even after they have promised | **0.81** | -0.08 |
| 4 | I have felt that I cannot communicate well with patients | **0.79** | 0.08 |
| 7 | I have felt uncertain about my effect on patients | **0.77** | 0.05 |
| 5 | I have had to repeat the same explanation to patients | **0.74** | -0.07 |
| 10 | Symptoms and treatment effects are obscure | **0.63** | 0.01 |
| 2 | I have had patients not participate in treatment when I tell them to | **0.58** | -0.02 |
| Factor 2: Nurses' role loss (Cronbach's α: 0.80) | | | |
| 21 | I have felt that I am not needed as a nurse by patients | -0.12 | **0.74** |
| 22 | Other staff members are more needed by patients than I am | 0.10 | **0.72** |
| 25 | I have lost confidence in my ability as a nurse owing to failure | -0.07 | **0.69** |
| 20 | I have felt that it is difficult to learn on my own | 0.10 | **0.62** |
| 24 | I have forgotten to greet patients as the days go by | 0.09 | **0.56** |
| 15 | I am anxious about my impaired judgment by the busyness of daily work | -0.10 | **0.53** |
| | Factors correlations | | |
| | Factor 1 | -- | |
| | Factor 2 | 0.29 | -- |

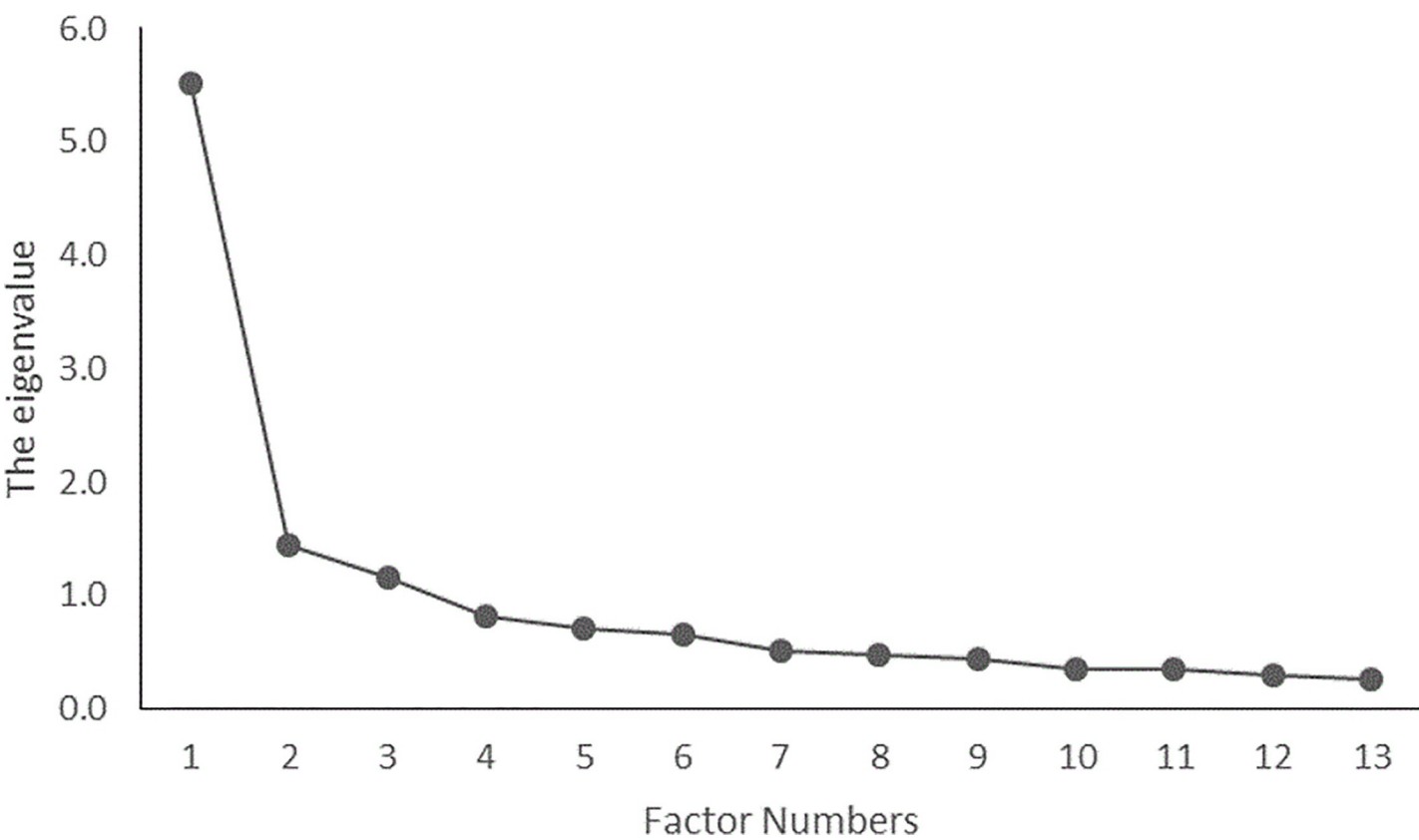

**Fig 1. Factor decay rate related to improved self-efficacy among psychiatric nurses.**

## Validity of factors

Regarding validity, the correlation between factor scores and GSES-J score were as follows: "Factors Related to Improved Self-efficacy (Total)," $r = 0.42$ ($p < .001$); "Positive reactions of patients," $r = .28$ ($p = .001$); "Ability to positively change nurse–patient relationship," $r = .44$ ($p < .001$); "Practicability of appropriate nursing," $r = .40$ ($p < .001$); "Factors Related to Decreased Self-efficacy (Total)," $r = -0.27$ ($p = .002$); "Uncertainty in psychiatric nursing," $r = -.05$ ($p = .55$); and "Nurses' role loss," $r = -.40$ ($p < .001$) (Table 5).

## Discussion

This study aimed to determine the factors related to psychiatric nurses' self-efficacy. Notably, these factors were partially different with respect to construct, compared to the factors in the previous study [9]. They were also partially similar to a previous study for general nurses [20].

### Factors that improve self-efficacy

The KMO measure of sampling adequacy exceeded 0.8 [21] and Bartlett's test was significant, indicating acceptable results. "Positive reactions of patients" included three items from the factor "the patients presenting a positive attitude" and one item from the factor "building a relationship of trust with the patients" in the previous study [9]. These four items reflect the meaning of the "Positive reactions of patients." "Positive reactions of patients" was partially similar in the factor construct in previous study. Additionally, "Ability to positively change

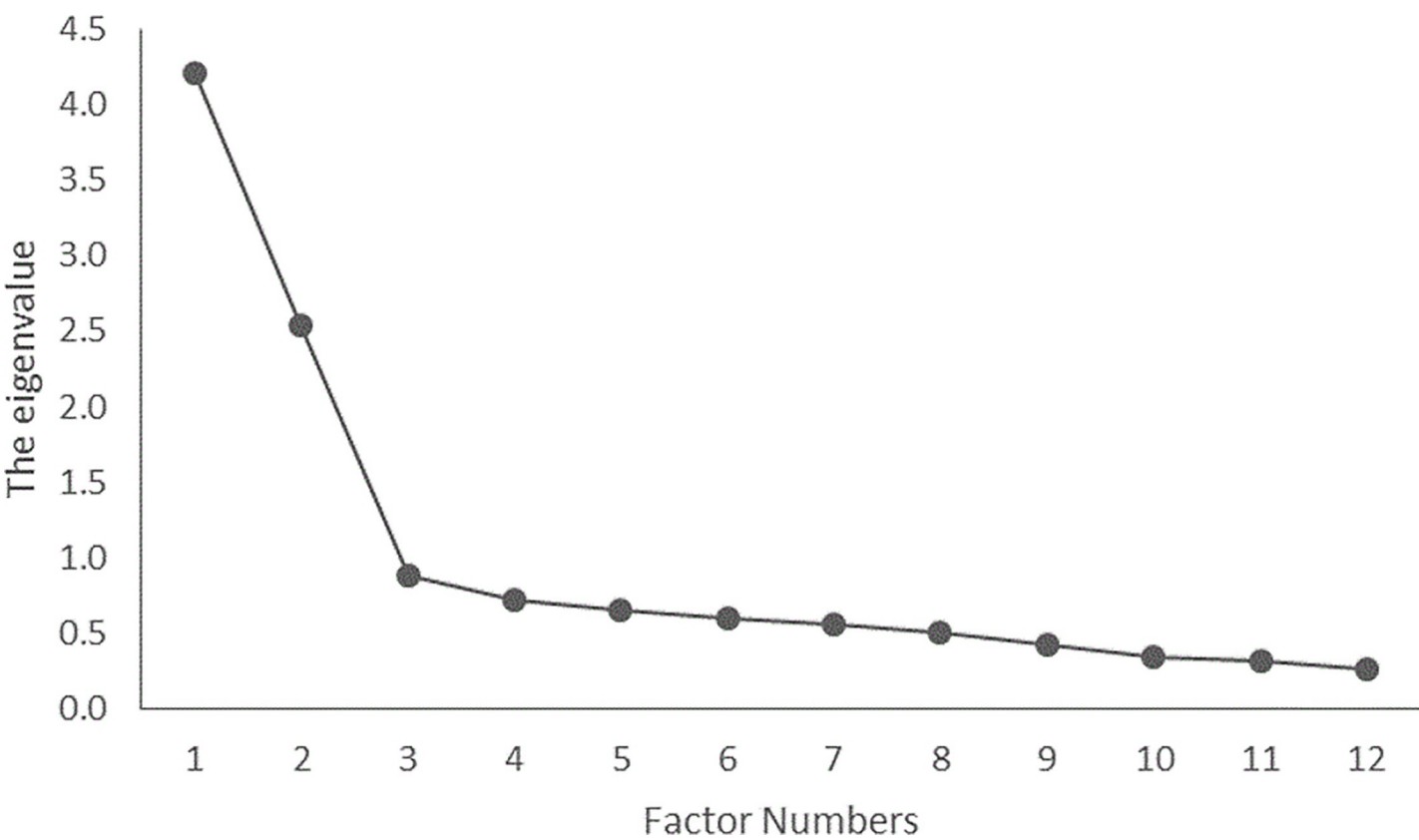

**Fig 2. Factor decay rate related to decreased self-efficacy among psychiatric nurses.**

nurse-patient relationship" included two items from the factor "the patients presenting a positive attitude," two items from the factor "building a relationship of trust with the patients," and one item of from the factor "building a relationship of trust with other nurses" [9]. This factor has implications for many factors from a previous study. In a study of general nurses [20], those with more positive work experiences reported high self-efficacy. Psychiatric nurses' self-efficacy may increase with positive experiences with the patients. Moderate correlations existed between "Positive reactions of patients" and "Ability to positively change nurse-patient relationship," which were similar concepts (Table 1). Interactions in the patient-nurse relationship may lead to improved self-efficacy among psychiatric nurses, because patient satisfaction is related to nurses' self-efficacy [22]. Perceived nursing care and satisfaction with care are strongly correlated in psychiatric care [23]. However, concerns have been raised that the lack

**Table 4. Internal consistency of factors related to self-efficacy among psychiatric nurses.**

|  | Cronbach's α coefficient |
|---|---|
| Factors Related to Improved Self-efficacy (Total) | 0.86 |
| Positive reactions of patients | 0.80 |
| Ability to positively change nurse-patient relationship | 0.84 |
| Practicability of appropriate nursing | 0.71 |
| Factors Related to Decreased Self-efficacy (Total) | 0.80 |
| Uncertainty in psychiatric nursing | 0.86 |
| Nurses' role loss | 0.80 |

**Table 5. The factors related to self-efficacy and the GSES-J correlations.**

| | GSES-J |
|---|---|
| Factors Related to Improved Self-efficacy (Total) | 0.42* |
| Positive reactions of patients | 0.28* |
| Ability to positively change nurse-patient relationship | 0.44* |
| Practicability of appropriate nursing | 0.40* |
| Factors Related to Decreased Self-efficacy (Total) | -0.27* |
| Uncertainty in psychiatric nursing | -0.05 |
| Nurses' role loss | -.398* |

*$p < 0.01$

of communication between patients and medical personnel may lead to a situation in which Japanese people are distanced from mental healthcare [24]; therefore, patients' satisfaction with mental health services may be low. Thus, improving this relationship might lead to improved self-efficacy among psychiatric nurses.

"Practicability of appropriate nursing" included two items from the factor "nursing judgment," one item from the factor "possibility of practical use in nursing," and one item from the factor "improvement of psychiatric symptoms" in the previous study [9]. This factor has implications for many factors from the previous study. However, this factor is also a new concept that has not been examined in the previous study [20]. Concerning "Practicability of appropriate nursing," inadequate communication between psychiatric patients and nurses may result in patients expressing anger toward nurses and lacking understanding and empathy [25]. Psychiatric nurses are stressed by their own low psychiatric nursing ability and communication with psychiatric patients [26]. Therefore, promoting nurses' ability to practice appropriate nursing may enhance their self-efficacy.

### Factors that decrease self-efficacy

The KMO measure of sampling adequacy exceeded 0.8 [21], and Bartlett's test was significant, indicating acceptable results. "Uncertainty in psychiatric nursing" included three items from the factor "uncertainty in caregiving" and three from the factor "lack of communication" in the previous study [9]. This factor was similar to those in the previous study. It was also a new concept not found in the previous study with general nurses [20]. Concerning the first factor —"Uncertainty in psychiatric nursing"—uncertainty leads to considerable anxiety [27], and researchers have been trying to understand the uncertainty of nursing [28]. Psychiatric nurses engage in nursing in uncertain situations where they have queries as to their nursing abilities and different attitudes towards nursing compared to non-psychiatric nurses [26]; therefore, psychiatric nurses' self-efficacy may be more influenced by nursing uncertainty than that of general nurses.

"Nurses' role loss" included two items from the factor "sense of loss regarding one's role as a nurse," one item from the factor "fluctuating view of nursing due to mistakes," one item from the factor "difficulty in bringing about self-improvement," one item from the factor "mechanical performance of nursing," and one item from the factor "sense of being too busy to work adequately" in the previous study [9]. This factor was a new concept not found in the previous study. As in the previous study with general nurses [20], nursing role was a factor of self-efficacy. Realization of a psychiatric nurse's own role may improve self-efficacy. Concerning "Nurses' role loss," as mentioned above, the lack of communication between healthcare

providers and the resulting distance between patients and nurses [24] might result in role loss and decreased self-efficacy among nurses.

## Examination of the factors' reliability

Concerning factor reliability, Cronbach's alpha should exceed .60, and scores greater than .95 indicate redundancy [29]; therefore, the internal consistencies of all the factors were considered acceptable.

## Examination of the factors' validity

Concerning validity, the correlations between the factors and GSES-J scores ranged from weak to moderate, except for "Uncertainty in psychiatric nursing." Due to the unique factors related to psychiatric nurses' self-efficacy, "Uncertainty in psychiatric nursing" may neither grasp psychiatric nurses' self-efficacy nor reflect strong self-efficacy. Therefore, it is necessary to re-select items based on the results of this study to create a robust scale.

## Study limitations

All 132 participants were nurses from public and private psychiatric hospitals; therefore, the possibility of selection and sampling bias cannot be excluded. This study utilized cross-sectional data; therefore, a longitudinal survey using a test-retest method is required. In this study, the factor structure was not confirmed. Confirmatory factor analysis requires more than 200 samples [30]. A confirmatory factor analysis with increased sample size is required to examine the validity of the factor structure. In addition, basic attributes such as marital status may affect self-efficacy. Future research should also gather information on marital status.

## Conclusions

In this study, three factors related to improved self-efficacy and two related to decreased self-efficacy among psychiatric nurses were extracted. These factors were partially different in terms of construct compared to the factors in a previous study [9]. The validity was quantitatively confirmed in 4 out of the 5 factors. Interventions based on these four factors may improve psychiatric nurses' self-efficacy; therefore, future planned interventions should examine these identified factors further in order to address ways to improve nurses' self-efficacy. However, some factors were weakly correlated; thus, unique facets of self-efficacy may have been measured as opposed to general self-efficacy clarifying the need to identify those unique factors. It is necessary to develop a scale that can evaluate psychiatric nurses' self-efficacy in more detail in the future.

## Supporting information

**S1 Data.**
(XLSX)

## Acknowledgments

We thank the psychiatric nurses for their cooperation and Yamaguchi University students who helped collect questionnaires and organize data. Furthermore, we thank Editage (www. editage.jp) for English-language editing.

## Author Contributions

**Conceptualization:** Hironori Yada, Hiroshi Abe, Ryo Odachi, Keiichiro Adachi.

**Data curation:** Hironori Yada.

**Formal analysis:** Hironori Yada.

**Funding acquisition:** Hironori Yada.

**Investigation:** Hironori Yada.

**Methodology:** Hironori Yada, Hiroshi Abe, Ryo Odachi, Keiichiro Adachi.

**Project administration:** Hironori Yada.

**Resources:** Hironori Yada.

**Software:** Hironori Yada.

**Supervision:** Hironori Yada.

**Validation:** Hironori Yada, Hiroshi Abe, Ryo Odachi, Keiichiro Adachi.

**Visualization:** Hironori Yada, Hiroshi Abe, Ryo Odachi, Keiichiro Adachi.

**Writing – original draft:** Hironori Yada, Hiroshi Abe, Ryo Odachi, Keiichiro Adachi.

**Writing – review & editing:** Hironori Yada, Hiroshi Abe, Ryo Odachi, Keiichiro Adachi.

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
