## [Decision Letter · Decision Letter 0]

2 Jan 2020

PONE-D-19-34276

Exploratory study of factors influencing self-efficacy among psychiatric nurses

PLOS ONE

Dear Hironori Yada,

Thank you for submitting your manuscript to PLOS ONE. After careful consideration, we feel that it has merit but does not fully meet PLOS ONE’s publication criteria as it currently stands. Therefore, we invite you to submit a revised version of the manuscript that addresses the points raised during the review process.

We would appreciate receiving your revised manuscript by 2/4/2020. To enhance the reproducibility of your results, we recommend that if applicable you deposit your laboratory protocols in protocols.io, where a protocol can be assigned its own identifier (DOI) such that it can be cited independently in the future. For instructions see: http://journals.plos.org/plosone/s/submission-guidelines#loc-laboratory-protocols

We look forward to receiving your revised manuscript.

Kind regards,

Karen-Leigh Edward

Academic Editor

PLOS ONE

Journal Requirements:

Reviewers' comments:

Reviewer's Responses to Questions

**Comments to the Author**

1. Is the manuscript technically sound, and do the data support the conclusions?

Reviewer #1: Partly

Reviewer #2: Partly

Reviewer #3: Partly

Reviewer #4: No

Reviewer #5: Yes

2. Has the statistical analysis been performed appropriately and rigorously? 

Reviewer #1: Yes

Reviewer #2: Yes

Reviewer #3: Yes

Reviewer #4: No

Reviewer #5: Yes

3. Have the authors made all data underlying the findings in their manuscript fully available?

Reviewer #1: No

Reviewer #2: No

Reviewer #3: No

Reviewer #4: No

Reviewer #5: No

4. Is the manuscript presented in an intelligible fashion and written in standard English?

Reviewer #1: Yes

Reviewer #2: Yes

Reviewer #3: No

Reviewer #4: No

Reviewer #5: Yes

5. Review Comments to the Author

Reviewer #1: The research idea is interesting. However, the authors need to provide clear description in some parts of the manuscript.

1. The title need editing as does reflect the aim of the study I do suggest to read "Exploration of the factors that influence self-efficacy among psychiatric nurses"

2. The abstract seems incomplete.

The background and methods lacks some key information

Results are well presented through need some editing

The discussion need to discuss the key findings and avoid repletion of result presentations

The conclusion should be drawn in relationship to the aim of the study

Refences need to updated.

The details are prrovide insitu (in the manuscript).

Reviewer #2: It seems that the present manuscript present the results of a study related to development and psychometric assessment of psychiatric nurses self efficacy, but title and abstract does not convey this issue to readers. So it is suggested to revise title and abstract to be more informative about the topic. It is better to add some numerical findings of study in abstract too.

Introduction does not explain the importance of present research sufficiently. The other point which might need authors' consideration is that the preliminary version of the developed scale was not assessed using face and content validity.

In extracting the factors and identifying the number of factors, scree plot might be useful. discussion does not cover all aspects of findings, it is too brief. also in reporting the the results it is better to report the variance of self efficacy explained by this scale.

Conclusion is not provided based on present findings (line 192-195).

Reviewer #3: This article focuses on the group “psychiatric nurses”, is a very interesting topic. I have some suggestions:

1. Please explain how to calculate the sample size.

2. Please clarify the inclusion and exclusion criteria.

3. Is the data collection completed in 2016? if so, does it can represent the current situation?

4. What is the marital status of the nurses? I think this could influence the self-efficacy among these nurses, more details about personal information is needed. It would be better to add a demographic table.

5. In line 69, what is the name of “The scale”? Does higher score mean better self-efficacy? It should be described in more detail.

6. The Measuring Instrument of GSES-J also should be described in more detail. Does higher score mean better self-efficacy?

7. InTable1, Item 2 appears twice, but the content is different. Please confirm it.

8. The logic and readability of the full text language need to be improved.

Reviewer #4: You mentioned that the purpose of the study was to identify the mental health of psychiatric nurses by exploring factors related to the self-efficacy of psychiatric nurses. However, there is a lack of logic about the relevance of self-efficiency and mental health of psychiatric nurses. The relationship between the mental health and self efficacy of psychiatric nurses should be described in the introduction part of manuscript in detail.

However, in the method and the results parts of the study, you implemented exploratory factor analysis to analyze the factors of the instrument consisting of 25 items that increase the sense of self-efficiency and 25 items that decrease the sense of self-efficacy. What is your purpose of the study?

Are you testing the reliability and validity of a tool that can measure the increase and decrease in psychiatric nurses' self-efficacy? I don’t know your study purposes.

There was no criteria for the sample size. 132 participants is not enough to make a exploratory factor analysis for 50 questions. Among the total 50 questions, only 25 questions remained. The deleting process of items was not described.

Moreover the correlation with the general self-efficiency scale was too low to show predictive validity.

Reviewer #5: Thank you for allowing me to review your study, Exploratory study of factors influencing self-efficacy among psychiatric nurses. It is important topic that needs further investigation because of the unique role of the psychiatric nurse. The study examined the responses from a survey consisting of 49 items used to explain self-efficacy. However, the GSES contained 23 items and it was not clear if this was also given to each participant. The abstracts states there were a total of 49 items.

The introduction needs to be revised beginning with the definition of Bandura’s self-efficacy and why this is in important concept for psychiatric nurses. Nursing is a stressful profession but the authors need to be more precise about how stress affects psychiatric nurses.

Some methodological issues- The authors state that the study was anonymous but the survey were collected by the nurse manger. It is unclear whether or not this person had access to the responses or who on their staff completed the survey. This design might have placed increased pressure on participants to complete the survey and needs to be explained in the paper how the researchers protected the data. In line 57 uses the term assistant nurse. The role of this person needs to be defined and included in the demographic table (table 1).

Table 1 should include the demographics of the sample and what type of facility the RNs worked in, e.g. academic center, community hospital, etc. This is an important table and the authors should consider what other items might need to be included.

Results-Surveys were completed from 147 but the results states 132. The authors need to explain this discrepancy.

An appendix with the survey would be helpful to the readers in analyzing the results.

In the statistical section there are some results regarding the analysis. This should be in the results section.

A figure with the factor loadings that were significant would help clarify the results. It is easy to become confused with the factor loading tables.

Discussion-The discussion should be revised to be stronger with the main outcome of the study –i.e. the important factors that impacted self-efficacy. This should be compared with other Japanese studies of nurses’ self-efficacy and how it is different, the same in this study.

The leading sentence in the discussion is confusing-This study aimed to clarify psychiatric nurses’ mental health by exploring the factors related to their self-efficacy. Do the authors really mean this study was designed to assess the mental health of psychiatric nurses?

The second sentence in the discussion should be expanded

Notably, the extracted factors were similar to the qualitatively extracted factors in previous studies. Explain what factors and what studies. This is probably the most important part of the paper and needs to very clear.

Clarification needed:

The number of missing answers in each item was 0–4, which we judged as small. (line 106).What does this mean?

This study have cross-sectional data; therefore, a longitudinal survey using a test-retest method is required (Line 188).

The authors need to correct all grammar errors and wrong tense of verbs-e.g. line 63 not . See above line 188 “have”.

Organisation for Economic Cooperation and Development countries-(line 188). Explain what is the purpose of this organization. Scientists reading this from other countries might not know. Spelling of organization-both in the text and references.

6. PLOS authors have the option to publish the peer review history of their article (what does this mean?). If published, this will include your full peer review and any attached files.

Reviewer #1: No

Reviewer #2: No

Reviewer #3: No

Reviewer #4: No

Reviewer #5: No

---

## [Author Response · Author response to Decision Letter 0]

14 Feb 2020

I have attached a response to each reviewer to the manuscript.

---

## [Editor Report · Decision Letter 1]

9 Mar 2020

Exploration of the factors related to self-efficacy among psychiatric nurses

PONE-D-19-34276R1

Dear Dr. Yada,

We are pleased to inform you that your manuscript has been judged scientifically suitable for publication and will be formally accepted for publication once it complies with all outstanding technical requirements.

With kind regards,

Karen-Leigh Edward

Academic Editor

PLOS ONE
---

## [Editor Report · Acceptance letter]

13 Mar 2020

PONE-D-19-34276R1 

Exploration of the factors related to self-efficacy among psychiatric nurses 

Dear Dr. Yada:

I am pleased to inform you that your manuscript has been deemed suitable for publication in PLOS ONE. Congratulations! Your manuscript is now with our production department. 

With kind regards,

on behalf of

Professor Karen-Leigh Edward 

Academic Editor

PLOS ONE